# Quinoa for the Brazilian Cerrado: Agronomic Characteristics of Elite Genotypes under Different Water Regimes

**DOI:** 10.3390/plants10081591

**Published:** 2021-08-02

**Authors:** Patrícia Carvalho da Silva, Walter Quadros Ribeiro Junior, Maria Lucrecia Gerosa Ramos, Sonia Maria Costa Celestino, Alberto do Nascimento Silva, Raphael Augusto das Chagas Noqueli Casari, Charles Cardoso Santana, Cristiane Andrea de Lima, Thomas Christopher Rhys Williams, Christina Cleo Vinson

**Affiliations:** 1Faculdade de Agronomia e Medicina Veterinária, Universidade de Brasília, Brasília 70910970, DF, Brazil; patriciacarvalhoagro@gmail.com (P.C.d.S.); lucreciaunb@gmail.com (M.L.G.R.); albertosilvaagro@gmail.com (A.d.N.S.); ccsantana.agro@gmail.com (C.C.S.); agro.cristiane@gmail.com (C.A.d.L.); ccvinson@unb.br (C.C.V.); 2Embrapa Cerrados, Empresa Brasileira de Pesquisa Agropecuária, Planaltina 73310970, DF, Brazil; sonia.celestino@embrapa.br; 3Instituto de Geociências, Universidade de Brasília, Brasília 70910970, DF, Brazil; casari.raphael@gmail.com; 4Departamento de Botânica, Instituto de Ciências Biológicas, Universidade de Brasília, Brasília 70910970, DF, Brazil; tcrwilliams@unb.br

**Keywords:** *Chenopodium quinoa*, water use efficiency, phenolic compounds, gas exchange

## Abstract

Quinoa stands out as an excellent crop in the Cerrado region for cultivation in the off-season or irrigated winter season. Here, we tested the effects of different water regimes on the agronomic characteristics, physiology, and grain quality of different elite quinoa genotypes under field conditions. The experiment was conducted under field conditions at Embrapa Cerrados (Planaltina, DF, Brazil). The experimental design was in randomized blocks, in a split-plot scheme, with four replications. The plots were composed of 18 quinoa genotypes and modified BRS Piabiru (the currently used genotype), and the split-plots were divided into 4 different water regimes. The following variables were evaluated: productivity and productivity per unit of applied water (PUAA), plant height, flavonoids, anthocyanins, gas exchange, chlorophyll, leaf proline, and relative water content. Our results showed that water regimes between 309 and 389 mm can be recommended for quinoa in the Cerrado region. CPAC6 and CPAC13 presented the highest yield and PUAA under high and intermediate WRs, and hence were the most suitable for winter growth under irrigation. CPAC17 is most suitable for off-season growth under rainfed conditions, as it presented the highest PUAA under the low WRs (247 and 150). CPAC9 stood out in terms of accumulation of flavonoids and anthocyanins in all WRs. Physiological analyses revealed different responses of the genotypes to water restriction, together with symptoms of stress under lower water regimes. Our study reinforces the importance of detailed analyses of the relationship between productivity, physiology, and water use when choosing genotypes for planting and harvest in different seasons.

## 1. Introduction

Crop development and yield are affected by different environmental factors, and water restriction is the most important constraint on agricultural yield [1,2]. This is a particular problem in the Brazilian Cerrado, which has a tropical climate with an average of 1500 mm of rain, but where approximately 90% of precipitation occurs during the rainy season (from October to April). The rainy season is followed by a dry season (from May to September), during which the relative humidity is low, the evaporation very high, and precipitation is rare. There are three harvest periods in the Brazilian Cerrado: (1) The main crop season, which occurs during the wet season from October to January; (2) The off-season crop, which is grown at the end of wet season without irrigation, is planted between the months of January to March [3,4], and is harvested in May during the dry season; and (3) the winter season crop, which is cultivated under irrigated conditions, with the crop being both planted (April to May) and harvested (August to September) during the dry season. Both the off-season and winter season require careful selection of genotypes for grain production; drought-tolerant genotypes (DT) should be selected for the off-season crop, and high productivity per unit of applied water (PUAA) genotypes are needed for winter crops as they are grown under irrigation. Obtaining genotypes that are better adapted to stressful edaphoclimatic conditions in order to resist periods with water deficiency whilst maintaining the highest possible productivity for each condition is therefore of great importance in plant breeding programs [5].

Quinoa (*Chenopodium quinoa* Willd.) is a pseudocereal rich in natural antioxidants, flavonoids, and anthocyanins [6,7], and these compounds may protect plants against biotic and abiotic stress [8]. Water stress increases leaf temperature, and reduces crop height, stomatal conductance, plant biomass, and yield [9,10].

Quinoa has been cultivated for millennia under conditions of low rainfall, as it has physiological and morphological strategies to overcome water deficit [11]. Moreover, this crop has been cultivated in different agroclimatic zones as it is well adapted to a variety of different environments due to its high genetic diversity [11]. Quinoa has mainly been cultivated in Argentina, Bolivia, Chile, Colombia, Ecuador, and Peru, though high productivity has also been observed when planted in Kenya as well as the Himalayas and northern plains of India [12].

In Brazil, research carried out in Embrapa Cerrados led to the selection of the genotype BRS Piabiru [3], which is the first cultivar in use for quinoa grain cultivation that is adapted to Cerrado conditions. Although planting quinoa during the main crop is not recommended due to the high water availability during the harvesting period (which can potentially result in panicle seed germination [3,5]), quinoa is recommended for growth during the off-season or irrigated winter season due to its high water use efficiency, drought tolerance, and adaptation to different environmental conditions.

Here, we therefore analysed growth, yield, and physiological and functional grain parameters, with the aim of identifying genotypes with potential for growth during the off-season and winter season for the Cerrado region.

## 2. Material and Methods

### 2.1. Experimental Area Characterization

The experiment was conducted between May and September 2017 at the experimental station of Embrapa Cerrados-Planaltina DF (Brazil), located between the geographic coordinates: 15°35′ S and 47°42′ W with an approximate altitude of 1200 m.

The climate of the region is tropical wet–dry (Aw), according to Köppen’s classification [13], with 2 well-defined seasons (dry and rainy). The average annual temperature is 21.3 °C and the mean annual rainfall 1400 mm. The experiment was conducted in winter due to the absence of rainfall during this period, which allowed for effective control of the amount of water applied. The long-term temperature and rainfall data from the last 20 years (1997 to 2017) are shown in Figure 1 and illustrate the annual dry season during the months of May to September. Temperature and rainfall data from the study period are in good agreement with long-term patterns (Figure 1).

The soil of the experimental area is classified as clayey Oxisol (Typic Haplustox) (Soil Survey Staff 2014), with a soft undulating relief and a clayey texture. The soil has the following characteristics in the 0–20 cm layer: pH (in water) = 5.71; Ca (mg kg^−1^) = 543.08; Mg (mg kg^−1^) = 103.32; K (mg kg^−1^) = 56; H + Al (mg kg^−1^) = 56.82; *p* (mg kg^−1^) = 22.87; organic matter (g kg^−1^) = 24.7.

### 2.2. Experimental Design

Tests with 980 selected genotypes were previously conducted in Cristalina-GO, and genotypes with an early cycle were selected; early cycles were defined as those of equal to or less than 110 days as this is the period for growth in the off-season or irrigated winter season. The experiment was repeated over 4 years; genotypes were sown in the off-season and winter of 2011, 2012, 2013, and 2014, and morphological characteristics were evaluated to obtain morphological descriptors of the genotypes. As a result, 18 genotypes with life cycle equal to or less than 110 days and with seeds larger than 2 mm in diameter were selected for use in this study: CPAC1, CPAC2, CPAC3, CPAC4, CPAC5, CPAC6, CPAC8, CPAC10, CPAC11, CPAC12, CPAC13, CPAC14, CPAC16, CPAC17, CPAC18, CPAC19, and CPAC20, together with the BRS Piabiru cultivar, which was modified by excluding high-cycle individuals.

The experimental design was in randomized blocks with split-plots with 4 replications. The plots were composed of the 18 quinoa genotypes and modified BRS Piabiru each separated by a distance of 1 m, and the split-plots were composed of 4 water regimes (WRs 150, 247, 389, and 480 mm). Seeds of the different genotypes of quinoa were sown manually under a no-tillage system. After soil analysis, fertilization was carried out in the furrows at a level of 400 kg ha^−1^ of the formula 04-30-16 (NPK). Sowing was done with a density of 0.7 g of seeds per linear meter and line spacing of 0.5 m. Nitrogen topdressing was carried out 30 days after seedling emergence at a dose of 100 kg ha^−1^ N in the form of urea. In order to avoid the competition of invasive plants, manual weeding was carried out.

The different water regimes were obtained using a sprinkler irrigator bar 40 m wide, connected to a spool with adjustable speed. Irrigation was performed homogeneously for the first 30 days after emergence in order to establish growth of all plants, with a total applied depth of 126 mm. After this period, the modified line source methodology was applied [14], using sprinklers with decreasing water flows from the central area to the end of the bar. The sprinklers were overlapped in order to promote a decreasing gradient of water from the central area to the edge of the bar.

Each water regime consisted of an experimental unit of 2.5 m^2^ in area within the plot (subplot), formed by 5 lines, with 0.5 m spacing. Over the stress treatment 10 irrigations were carried out, and the accumulated uniform irrigation plus the variable irrigations gave a total of 150, 247, 389, and 480 mm for the 4 water regimes. The highest level of irrigation was determined as described in the irrigation monitoring program in the Cerrado [15], through replacement of evapotranspiration using agrometeorological indicators of the region, the soil type, the date of full emergence of plants, and wheat as a reference. Although our experiment was carried out with quinoa, we used wheat as a reference crop for irrigation, understanding that it has similar water consumption. Irrigation was carried out approximately every 5 days, according to the climatic conditions and crop phenological phase. To measure the water depth applied in each irrigation 2 rows of collectors parallel to the irrigation line were set up.

Thermal images were acquired using a thermal infrared camera (FLIR^®^ T420, FLIR Systems, Wilsonville, OR, USA) with the following characteristics: thermal spatial resolution of 320 × 240 pixels, spectral response of 7.5 to 13 µm, thermal sensitivity of pixels from 0.045 °C to 30 °C, and temperature accuracy of ±2. The camera was mounted on an unmanned aerial vehicle (XFly, O X800, Bauru, SP, Brazil) at a height of 60 m from the canopy and the images were performed at the same time and day as the physiological variables. For the processing of images and to obtain temperature data in the canopies of quinoa plants, the QGIS software was used [16]. The RGB (red, green and blue) and thermal images were geo-referenced for the generation of an orthomosaic and later classified to select only the areas of plant canopies in the useful parcel, avoiding picking up exposed soil.

### 2.3. Variables Analyzed

#### 2.3.1. Grain Analysis

Quinoa grain productivity and productivity per unit of applied water (PUAA) were evaluated for all studied genotypes under 4 water regimes (WRs 150, 247, 389 and 480 mm). The panicles were harvested by hand, and later placed in a Wintersteiger plot harvester. In each subplot an area of 2.5 m^2^ was harvested. Grain moisture was corrected to 13% and the results were analysed as ton ha^−1^. The productivity per unit of applied water (PUAA) was determined by the ratio between the grain yield and the amount of water applied during the whole crop cycle, analysed in kg ha^−1^ mm^−1^. Productivity per unit of applied water (PUAA) was also evaluated, and calculated using the following equation: PUAA=Prod/LTD
where

PUAA: Productivity per unit of water applied (kg ha mm^−1^);

Prod: Grain productivity, in kg ha^−1^; and

LTD: Total irrigation depth available referring to the amount of water applied during the crop cycle, in mm.

The weight of 1000 grains was evaluated for 4 genotypes (CPAC4, CPAC11, CPAC19, and the early BRS Piabiru). The weight of 1000 grains was determined according to the method described by [17], in which 100 grains from each subplot are weighed using a semi-analytical balance and the result obtained is multiplied by 10.

The functional quality of the grains was evaluated for all studied genotypes. Functional quality was evaluated by quantification of flavonoids and total anthocyanins and flavonoids [18]. The grains were ground; subsequently, 5 g was weighed and the anthocyanin and flavonoids from the grains were extracted with 30 mL of a solution of ethyl alcohol and 1.5 N HCl. After homogenization for 1 min, the volume was made up to 50 mL in volumetric flasks. This mixture remained at 8 °C for 16 h. After filtration, absorbances (Abs) were taken at 535 nm for anthocyanin and 374 nm for flavonoids. The results were expressed in mg/100 g of quinoa grains. The contents of flavonoids 9 factor 76.6) and of anthocyanins (factor 98.2) were calculated according to the formulas: Dilution Factor (DF) = (100 × volume (mL))/weight (g)); Content of flavonoids/anthocyanins = (Abs × DF)/factor. Results were expressed in mg/100 g of quinoa grains.

#### 2.3.2. Physiological Analysis

During the flowering period of the crops, 60 days after emergence the following variables were evaluated: gas exchange (photosynthesis; stomatal conductance, internal CO_2_ concentration in the leaves, and transpiration), leaf chlorophyll (*a*, *b*, total), chlorophyl fluorescence, proline content, relative water content, and plant height. These variables were analysed for 4 genotypes (CPAC4, CPAC11, CPAC19, and the early BRS Piabiru), with exception of plant height, which was analysed in all genotypes. Plant height was measured from the base of the stem to the peak of its inflorescence using a measuring tape. Five plants per subplot were evaluated.

Gas exchange was measured using an IRGA infrared gas analyzer model LI-6400XT (LI-COR, Inc., Lincoln, NE, USA) between 08:00 and 11:00 on the youngest fully expanded leaves. Ambient air temperature and relative humidity were used and the CO_2_ concentration in the chamber was maintained at 400 µmol mol^−1^ together with a saturating photosynthetically active photon flux density (PPFD) of 2000 μmol m^−^^2^ s^−1^. To determine light saturation, a PPFD curve of 0, 20, 60, 100, 250, 500, 1000, 2000, 2250, 2500, and 3000 μmol m^−2^ s^−1^ was obtained under ambient CO_2_. Maximum fluorescence (F’m) and basal fluorescence (F’0) were evaluated for light-adapted leaves using the portable open flow gas exchange system (IRGA-LI-6400XT; LI-COR Inc., Lincoln, NE, USA). The effective quantum yield of Photosystem II is given by, F’v/F’m = (Fm’ − F0′)/Fm’, according to [19].

The chlorophyll index (*a*, *b*, *a + b* and *a/b*) was evaluated in the morning with the aid of a digital ChlorofiLOG, model CFL 1030, from Falker in the youngest fully expanded leaves, and 10 readings were taken for each treatment.

The relative water content (RWC) was determined according to the methodology of [20], using leaf discs of 1 cm in diameter obtained from leaves collected predawn between 03:00 and 05:00. The discs were immediately weighed after collection to determine the fresh mass (FM). The turgid mass (TM) of the leaf discs was obtained by hydration in distilled water for 24 h and the dry mass (DM) was obtained after drying in a forced air oven at 60 °C. The RWC was calculated according to the formula.

RWC = ((FM − DM)/(TM − DM)) × 100%,

The proline content was determined according to [21]. Completely expanded young leaves were collected between 13:30 and 15:00. Samples were extracted by grinding 0.5 g of fresh leaf material in 10 mL of 3% sulfosalicylic acid. Then, 500 µL of the extract of each sample was placed in a test tube containing 2.75 mL of distilled water, with the addition of 2 mL of acid ninhydrin, 2 mL of glacial acetic acid, and 100 microliters of glycine (126 mM). Acid ninhydrin was prepared by warming 1.25 g in 30 mL glacial acetic acid and 20 mL 6 M phosphoric acid. Then, the samples were kept for 1 h in a water bath at 100 °C [21]. After cooling the samples by immersion in an ice bath the optical density was evaluated at 515 nm using a Pharmacia Ultrospect III spectrophotometer. The absorbance obtained was compared with the standard proline curve and the results expressed as µmol of proline/g of fresh mass (FM).

### 2.4. Statistical Analysis

Data were subjected to analysis of variance at 5% probability by the F test and the comparison of means was carried out using the Scott–Knott test and Tukey’s test. The statistical model was adjusted using the Proc Mixed from SAS [22] using the restricted maximum likelihood method. For non-significant interactions, factors were analysed separately (genotype and water regime). Figures were produced using the software Sigma Plot, version 10 [23].

## 3. Results

### 3.1. Productivity

In general, the genotypes responded differently to water regimes, reinforcing the idea that the productive potential, productivity per unit of applied water (PUAA), and responsiveness to irrigation depend on the genotype (Table 1 and Table 2). We found a significant interaction between quinoa genotypes and WRs for productivity (Table 1). For most quinoa genotypes, the yield was reduced with WR 247, with exception of CPAC3, CPAC8, CPAC9, CPAC14, and CPAC19 which had reduced yield with WR 389 mm.

Under the highest WR (480 mm) all genotypes obtained similarly high yields, meaning that the genotypes could not be distinguished. The only exception was CPAC11 (a dwarf genotype), which presented significantly lower productivity (Table 1) and the lowest PUAA (Table 2). However, in the intermediate WR (389 mm), there were major differences among genotypes, with the genotypes CPAC6, CPAC13, and CPAC18 standing out as having the highest productivity (Table 1). Under the water deficit regimes there was a clearer distinction between the genotypes, which were divided into three groups: high, medium, and low productivity. Under the 247 mm WR, with a reduction of 49% in applied water, the genotypes presented a high productivity of between 2.53 and 6.26 t ha^−1^. On the other hand, under the extremely low water regime (150 mm), the genotypes CPAC6, CPAC10, CPAC11, CPAC12, CPAC13, CPAC16, and CPAC17 were more productive. The genotypes CPAC3, CPAC8, CPAC9, CPAC14, and CPAC19 were more sensitive to mild water deficit (WR 389), showing reduced productivity under this water regime, whilst CPA11 showed the opposite response, with increased productivity with 91 mm less water applied (Table 1).

### 3.2. Productivity Per Unit of Applied Water (PUAA)

In general, the lowest PUAA was found for WR 480, with only the genotypes CPAC8, CPAC16, CPAC19, and CPAC20 presenting similar PUAA at 480 mm and under deficit conditions (380 and 247 mm, Table 2). The highest PUAA occurred between 247 and 389 mm WR for most quinoa genotypes (Table 2). Comparing WR 480 mm with the intermediate 389 mm and 247 mm WRs, there was a reduction of 36 and 49% in volume of total water applied and an increase by 23 and 21% in PUAA, respectively (Table 2). The genotypes with the highest PUAA in 247 mm were CPAC3, CPAC6, CPAC9, and CPAC12, and with the exception of CPAC6, the other genotypes had low PUAA in the other WRs (Table 2).

Under the low water regime of 150 mm the PUAA for CPAC17 was significantly higher than the other genotypes, and importantly, this variable was similar under intermediate and low water regimes, meaning that regardless of the water regime its PUAA did not alter. Although CPAC17 had the highest productivity under WR 150 mm (3.64 ton ha^−1^) compared to all studied genotypes (Table 1), it had greater productivity under intermediate (8.44 ton ha^−1^) and high (8.92 ton ha^−1^) WRs. Under the lowest WR (128 mm), however, overall efficiency was very low as the stress was very high.

### 3.3. Thousand Grain Weight

The weight of 1000 grains (TGW) of quinoa was evaluated, and there was no significant interaction between the factors. The effect of the water regimes on TGW is shown in Appendix A), where there is an increase in TGW up to WR 389 mm and from this level, there was no increase in grain weight with the increase in water availability. The CPAC11 and early Piabiru genotypes showed better results, and CPAC19 had lower grain weights compared to other genotypes. In this study, a TGW of 2.35 to 2.84 was obtained, depending on the genotype (Appendix A).

### 3.4. Functional Quality

Our study shows a significant interaction between quinoa genotypes and water regimes and the abundance of flavonoids (Table 3) and total anthocyanins (Table 4). For flavonoid contents, BRS Piabiru and CPAC2 did not respond to all WRs, but showed lower values than CPAC9, which was influenced by WRs. For anthocyanin contents, CPAC1, CPAC11, CPAC16, CPAC18, CPAC19, and CPAC20 presented similar values in all WRs, and these genotypes had lower values than CPAC9.

The accumulation of flavonoids in CPAC9 was greater than in BRS Piabiru and all other genotypes in all water regimes, reaching approximately double the concentration of flavonoids in most genotypes in both higher and lower water regimes (Table 3).

Similar to that observed for flavonoid accumulation, CPAC9 also accumulated higher anthocyanin content and had greater concentrations than BRS Piabiru and other genotypes under all water regimes, accumulating approximately double the concentration of anthocyanins (Table 4). In addition, CPAC9 was among the genotypes with the highest productivity under both high (8.21 ton ha^−1^) and intermediate (7.01 ton ha^−1^) WRs (Table 1).

### 3.5. Plant Height

Figure 2A shows quinoa plant height in response to water regimes. In the higher water regimes (389 and 480 mm), the plants presented greater heights (1.50 and 1.51 m) than the intermediate (247 mm) and low (150 mm) water regimes, which presented plants with lower heights of 1.40 and 1.17 m, respectively (Figure 2B). Severe water deficit (128 mm) reduced plant height by 22%. Comparing among genotypes, genotype CPAC11 was significantly shorter than all others, with an average height of 0.94 m (Figure 2B).

### 3.6. Canopy Temperature

For the leaf temperature variable, no interaction between the factors was obtained and no significant difference was observed between the genotypes. Thus, the effect of water regimes on the temperature of the plants was subjected to regression analysis. This analysis showed a significant relationship between water availability and leaf temperature (*p* < 0.05), with the maximum temperature value (35.1 °C) for the lowest WR (150 mm) and the minimum (25.5 °C) in WR 480 mm (Appendix A), demonstrating that under severe drought the canopy of quinoa genotypes showed an increase in temperature of 9.6 °C.

### 3.7. Leaf Gas Exchange

The stomatal conductance and transpiration, as expected, were higher in the high-water regime and decreased as water restriction occurred, with a reduction of 94, 88, 92, and 91 % for the genotypes CPAC4, CPAC11, and CPAC19 and the BRS Piabiru precoce, respectively (Table 5). The different water conditions affected the four analysed genotypes differently, and CPAC9 had the highest stomatal conductance in the intermediate WR (247 mm), which represents off-season conditions. The CPAC11 genotype was more efficient in maintaining stomatal conductance and transpiration, as it had lower values in the high water regimes but little decrease in the intermediate WR (Table 5).

We detected increased internal carbon (Ci) under the high water regimes and decreased Ci under severe stress, which agrees with the stomatal conductance and transpiration data as the absorption of CO_2_ is directly related to the degree of stomatal opening. (Table 5). Similarly, photosynthesis of the four genotypes decreased gradually with reduced water availability. In the comparison between the highest (480 mm) and the lowest (150 mm) WR, reductions in photosynthetic rates of 83, 70, 74, and 75% were observed for the genotypes CPAC4, CPAC11, CPAC19, and BRS Piabiru precoce, respectively. An interesting pattern was observed for net photosynthesis, as under the high WR there were two groups of genotypes. CPAC19 and the BRS Piabiru obtained high values whilst CPAC4 and CPAC11 obtained lower values; however, their values were similar under intermediate WR and then suffered a drastic reduction in the severe WR. Ci proved to be the least sensitive parameter and could only distinguish the genotypes under the most severe water stress regime (WR 150).

### 3.8. Effective Quantum Yield of Photosystem II and Chlorophyll Index

For the F’v/F’m ratio, the factors varied independently, and no significant difference was observed between the genotypes. The effect of water regimes on the F’v/F’m ratio of quinoa plants was therefore analysed using regression. This analysis indicated a significant relationship between water availability and F’v/F’m (*p* < 0.05), with a maximum value (0.52) of F’v/F’m in the highest WR (480 mm) and the minimum (0.43) in WR 150 mm (Appendix A), indicating that the efficiency of PSII was significantly reduced when the plants were submitted to the severe stress of WR150 mm. In contrast, chlorophyll index was not affected by the water regimes, but there were differences between the genotypes, with CPAC19 having the greatest total chlorophyll index and CPAC11 the lowest principally due to differences in chlorophyll a. CPAC19 also stood our as having the lowest chlorophyll a to b ratio (Appendix A).

### 3.9. Leaf Proline and Relative Water Content

There was a significant interaction between the quinoa genotypes and the water regimes for leaf proline content (Appendix A). However, in general, proline concentrations were relatively insensitive to water restriction, as concentrations only increased under severe stress (Appendix A). We detected no effects of genotype on leaf relative water content (RWC). Regression analysis revealed a significant relationship between water regime and RWC and a maximal RWC of 81% at WR 480 mm and minimal RWC of 58% at WR 150 mm (Appendix A). The 19% reduction in water application of the WR 389 mm resulted in a minimal reduction in RWC (Appendix A).

## 4. Discussion

### 4.1. Productivity and Productivity Per Unit of Applied Water (PUAA)

Irrigation water use efficiency refers to the yield obtained per unit of applied water [24] and is a fundamental physiological parameter that indicates the ability of crops to conserve water in a region under water stress due to drought resistance and high potential productivity [25]. In our study, the low WR of 150 mm resulted in lower PUAA because under severe water restriction the quinoa genotypes cannot express their productive potential, whilst at high WRs (above 480 mm) plants also had a lower PUAA due to an inability to absorb all supplied water and potentially an intolerance to excess water (Table 2). Under the high WR, genotypes showed high productivity; however, the grain dry matter per unit of applied water was low, indicating that there was no consistent relationship between crop yield and PUAA for this WR (Table 1 and Table 2). Thus, the 389 mm and 247 mm WRs showed the highest PUAA, but the highest productivity was observed under the WR 480 mm and 389 mm. Thus, WR 389 mm can be indicated for cultivating quinoa under an irrigated system in the Cerrado, as there is a trade-off in the relationship between productivity and water saving.

Quinoa plants can control the relationship between photosynthetic rate and transpiration, even with low leaf water potentials [26], and by limiting transpiration and inducing stomatal closure, they can increase PUAA and influence productivity under water stress [27]. In our study, with a 49% reduction in water applied throughout the crop cycle there was a 42% average yield loss over all genotypes (Table 1), indicating both drought tolerance and efficient water use, as this grain yield was obtained using half the water normally needed to meet the demands of the crop. Quinoa seeds can be obtained when little water is available in the vegetative stage, producing an average of 1.2 to 2.0 t ha^−1^ with half the required irrigation [28,29]. On the other hand, when a low irrigation strategy was used during all phenological stages this resulted in a 75% reduction in seed yield of the quinoa cultivar ‘Belen 2000’ [28,29].

Our work obtained higher yield values than others reported in the literature, and under all WRs the genotypes with high productivity were CPAC13, CPAC6, CPAC3, CPAC12, and CPAC17 and the genotypes with lower yield potential were CPAC19, CPAC11, and CPAC14 [30,31]. Under higher WR (480 mm) the genotypes did not differ, with the exception of CPAC11, which presented the lowest productivity and low PUAA; however, CPAC11 was also the only dwarf material used in this study (see below) (Figure 2, Table 1 and Table 2). For the 150 mm WR the CPAC17 genotype was superior to the other genotypes, and whilst productivity was altered there were no changes in efficiency between WR 150 and WR 389 mm (Table 1), meaning that it is a suitable genotype for use in situations with limited water availability such as the off-season.

Under high and intermediate water regimes, the highest PUAA was observed for CPAC3, CPAC6, and CPAC12 between WR 480 and WR 247 mm (Table 2) and considering that they were amongst the genotypes with highest productivity, these genotypes are suggested for the winter season. Specifically, CPAC6 exhibited reduced productivity only under the 247 mm WR (Table 1) and presented the highest PUAA of 26.7 kg ha^−1^ mm^−1^. This value is 32% higher than the PUAA of WR 480 mm (Table 2). CPAC13 presented higher productive potential and productivity than BRS Piabiru under a moderate water regime (389 mm), with 9.73 t ha^−1^ for CPAC13 and 8.14 t ha^−1^ for BRS Piabiru, respectively.

### 4.2. The Effects of Water Regime and Genotype on Grain Quality Indicators

In addition to productivity indicators of grain quality such as the concentrations of flavonoids and anthocyanins and 1000-grain weight should also be taken into account when selecting genotypes (Table 3 and Table 4, Appendix A). Our results show that the accumulation of flavonoids and anthocyanins in quinoa plants was more influenced by genotype than by the WRs. In particular, CPAC9 accumulated these compounds under both higher and lower water regimes and accumulated nearly double the concentration of the other genotypes under all WRs. In addition, CPAC9 is among the genotypes with greater productivity and PUAA under high and intermediate WRs (Table 1 and Table 2). The authors of [32], when studying the levels of flavonoids, phenolic acids, and betaines in the Andean grains of quinoa, kaniwa, and kiwicha, found flavonoid contents ranging from 36.2 to 144.3 mg/100 g, which are similar than those found here. In other crops such as peanuts, depending on the genotype and drought period, concentrations of phenolic compounds in seeds may be between 60 and 220 mg/100 g [33]. Further studies focusing on the biosynthesis of phenolic compounds and oxidation processes under water stress will provide more information on the genotypic variation of phenolic content in grains [34]. Grain weight is also affected by water restriction. With a water supply of 150 mm, there was a 14% decrease in TGW for the four genotypes analysed, similar to a previous study where TGW in irrigated plants was significantly higher (5.5 g) than in rainfed plants (4.2 g) [28,29]. Indeed, when water stress is applied during the grain filling period, it generally reduces the grain yield, the number of grains per plant, and the individual weight of the grains [35].

### 4.3. Low Water Availability Leads to Reduced Plant Height

The heights of the quinoa genotypes in this study ranged from 0.99 to 1.53 m, which are superior to those in literature [36]. Our results support previous studies demonstrating that a reduction in irrigation resulted in a significant decrease in plant height by 0.55 and 0.80 m for Bolivian quinoa [37]. Here, we detected a decrease in cultivated quinoa plant height under 70% irrigation deficit in relation to control plants, similar to that observed by [38]. The reduction of plant size and leaf area under stress conditions is directly related to a decline in dry mass when compared to plants maintained under adequate soil water potential conditions [39]. The low water potential of the soil limits water absorption capacity, and this quickly suppresses the rate of cell expansion and division of growing tissues [40]. Moreover, under severe water deficit stomata close and the consequent reduction of photosynthesis results in a decrease in dry mass production [39]. Plant height can also be used as a criterion to determine the susceptibility of quinoa genotypes to drought, as the longer the cycle, the larger the plant. Short-to-medium cycle cultivars are desirable, and would provide the possibility of more crops per year in irrigated systems [2] and enable off-season and winter planting. A shorter crop cycle also represents a method of escape from water stress under Cerrado conditions, besides contributing to the definition of the sowing season, such that the grain maturation occurs when the humidity is reduced, thus avoiding seed deterioration [2]. Amongst the studied genotypes, CPAC11 was significantly shorter. Whilst this is ideal for avoiding lodging, this genotype also showed the lowest productivity in most WRs under the planting density we employed. This may be related to reduced cell expansion, which results in a reduction in leaf area and, consequently, less photoassimilates for translocation to the grain [41]. In addition, water deficit affects carbohydrate utilization, altering the efficiency with which photoassimilates are used in the growth and development of new plant organs [41]. This genotype (CPAC11), however, may not have expressed its productive potential considering that it is the only dwarf material used in this study (Figure 2) and may need a higher planting density per square meter.

### 4.4. Physiological Parameters

Abiotic stresses generally result in reduced rates of photosynthesis and transpiration that can ultimately contribute to reduced growth and productivity [41], making assessment of physiological parameters in the field an important tool in screening for stress tolerance. For this reason, we analysed a number of physiological parameters in a subset of the quinoa genotypes. Remote sensing of leaf temperature by thermal imaging can be a reliable way to detect changes in the physiological state of plants in response to different biotic and abiotic stresses [42]. Indeed, canopy temperature has been used successfully in breeding programs for drought-prone environments [43,44], as genotypes that maintain transpiration will tend to have lower canopy temperature than other genotypes under the same environmental conditions in the field [45]. Here, reduced water availability led to increased canopy temperature, reflecting the stomatal closure and reduced transpiration that we detected using gas exchange analysis. However, we were not able to detect differences between the genotypes using this parameter (Appendix A), despite the fact that differences in transpiration were detected between genotypes at each WR (Table 5). It may be the case that larger differences in transpiration between genotypes are required to result in alterations in leaf temperature that can be measured using this technique.

Stomatal closure is a common response to water restriction where water deficit affects guard cell turgidity and stomatal aperture, resulting in decreases in the rates of transpiration and assimilation of CO_2_ and increased leaf temperature [46,47,48]. Gas exchange parameters such as net photosynthesis, transpiration and stomatal conductance are therefore sensitive indicators of water deficit in plants that are useful in the evaluation of genotypes adapted for growing in environments with limited water availability. Our data are consistent with previous studies, since the photosynthetic rate presented a behaviour similar to that observed for stomatal conductance and transpiration, reflecting the opening and closing of stomata under different water conditions [49], and quinoa is known to suffer both stomatal and mesophyll limitations under drought stress [50] These results also suggest that the reduction in photosynthetic rates in the most stressed water regimes are related to partial stomatal closure and a consequent reduction in CO_2_ assimilation (Table 5). Water stress is considered to be severe when stomatal conductance values are below 0.1 mol.m^−2^ s^−1^ [47], and therefore in this study all genotypes were under severe stress in the 150 mm WR (Table 5). Similar values for stomatal conductance and effects of water restriction in quinoa have been reported elsewhere; for example stomatal conductance decreased from 0.213 mol.m^−2^ s^−1^ under irrigation to 0.091 mol.m^−2^ s^−1^ under water deficit in one study [51] whilst an investigation of 10 genotypes in the field without irrigation that received around 160 mm of water reported values between 1.0 and 0.18 mol.m^−2^ s^−1^ and a clear relationship between maintenance of stomatal conductance and higher photosynthetic rate [52].

Analysis of gas exchange also revealed differences in the responses of the genotypes to alterations in water availability and relationships with productivity. CPAC4 and CPAC11 appear most sensitive to water restriction as they presented the lowest net photosynthesis under the 150 mm and 247 mm regimes. Interestingly though, whilst CPAC19 showed greater photosynthesis under all water regimes, this was not reflected in greater productivity, indicating that other factors, such as plant morphology and the capacity to use photoassimilates for grain filling also has an important impact. This is also seen in the relationship between water regime, photosynthesis and productivity for individual genotypes; for example, the increased photosynthesis shown by CPAC4 between WR389 and WR480 did not translate into greater productivity, reinforcing the importance of parameters such as PUAA for selecting genotypes under irrigated conditions (Table 1, Table 2 and Table 5).

Gas exchange measurements may not always be able to detect the deleterious effects of water restriction on chloroplast function parameters such as the effective quantum yield of photosystem II potentially useful tools [53]. Here we detected decreased F’v/F’m with low water availability (WR 150 mm, Appendix A); this decrease in plants stressed by drought, in comparison with well-hydrated plants, is mainly due to the lack of CO_2_ inside the leaf and it is under this WR that we detected large decreases in Ci for all genotypes (Table 5). However, this parameter responded little to the median level of stress (389 mm), despite the fact that this WR provoked changes in several gas exchange parameters (Table 5), and furthermore we did not detect any differences between the genotypes (Appendix A). Indeed, measurements of chlorophyll a fluorescence tend to have low sensitivity to mild stresses, for example, 18 days of suspension of irrigation were required to reduce Fv/Fm in two greenhouse-grown quinoa genotypes [53]. In contrast to F’v/F’m, chlorophyll indices proved to be unaffected by WR but showed differences between genotypes (Appendix A), and lack of an effect of drought and flooding on chlorophyll in quinoa has previously been reported [52]. Despite the lack of an effect of stress on chlorophyll, these indices may prove useful for selection of genotypes due to the fact that chlorophyll abundance is typically positively related to photosynthetic potential and productivity [54]. In this sense of the four genotypes analysed CPAC19 stood out due to greater total chlorophyll and a lower chlorophyll a:b ratio that may indicate greater light absorption capacity by photosystem II [55].

A lack of water in the soil increases the risk of the rate of transpiration exceeding the rate of water absorption and transport, resulting in a situation of water deficit. Partial stomatal closure can reduced transpiration, but under water stress, plants often also accumulate compatible solutes or osmoprotectors including proline, glycine, betaine, and sugars [56]. The accumulation of compatible solutes reduces cellular osmotic potential, thereby permitting water absorption and maintaining turgor pressure and physiological processes [57]. The accumulation of proline may therefore be an important characteristic for the selection of drought-tolerant plants [58] and indeed seed or leaf treatment of quinoa plants with free proline can increase growth under water stress [59] whilst a number of studies have connected compatible proline accumulation with drought and salt tolerance in this species [49]. However, whilst we detected increased proline concentrations in quinoa in response to water stress, this only occurred under the most severe water regimes, meaning that it could not be used to discriminate between the genotypes under water regimes that reflect Cerrado conditions (Appendix A). Despite morphological alterations, stomatal control and osmotic adjustment water restriction may eventually affect leaf water status. Leaf relative water content (RWC) can therefore be used to indicate the balance between water supply and transpiration [60], and in the case of F’v/F’m, we did not detect differences between the genotypes for this parameter (Appendix A). However, RWC did serve to indicate the degree of stress to which the plants were subject, as the RWC values detected below 389 mm correspond to those associated with the beginning of wilting (Appendix A, [20] and are similar to those observed in greenhouse grown plants during suspension of irrigation [53].

## 5. Conclusions

Through experiments performed under different water regimes here we have shown that quinoa has excellent potential for planting as an off-season and winter crop in the Cerrado region. Several genotypes presented advantages in relation to the currently used BRS Piabiru genotype; the choice of genotype will depend on farming practices, nutritional content, and weather conditions. CPAC13 and CPAC6 are particularly suited to growth as a winter crop under irrigated conditions, and CPAC17 under off-season rain-fed conditions, whilst CPAC9 appears advantageous in terms of phenolic compounds in the grains. The accumulation of flavonoids and anthocyanins in quinoa genotypes was more influenced by quinoa genotype than by the WRs. Analysis of physiological parameters provided information regarding the mechanisms involved in stress tolerance in different quinoa genotypes, which is essential if we are to develop strategies to maintain or increase plant productivity in environments with water limitation. The results of this work show that the water regimes for quinoa can be reduced without a significant reduction in grain yield. This increase in dry matter accumulation efficiency per unit of water applied in quinoa means it is a crop that can be cultivated under Cerrado conditions, for both the off-season and winter season, under relatively low levels of irrigation whilst obtaining high yields. This fact, coupled with proper water management this can result in higher yield per area, which is desirable for areas under irrigated cultivation where irrigation is a costly practice.

## Figures and Tables

**Figure 1 plants-10-01591-f001:**
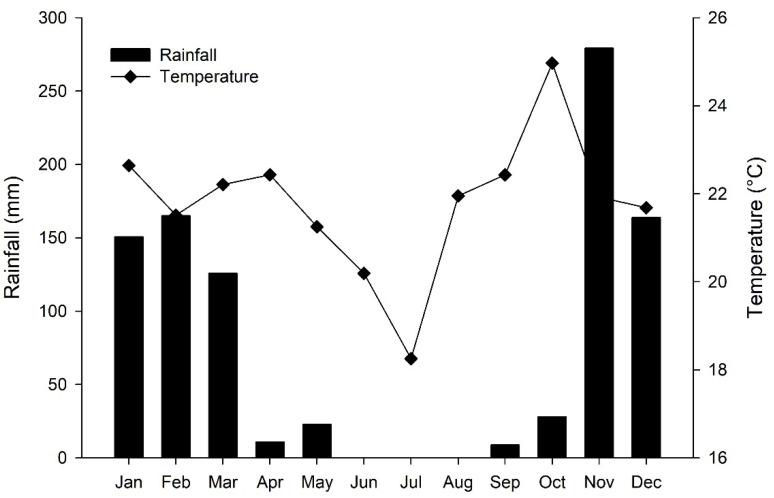
Mean rainfall and temperature in long-term data from the last 20 years (1997–2017).

**Figure 2 plants-10-01591-f002:**
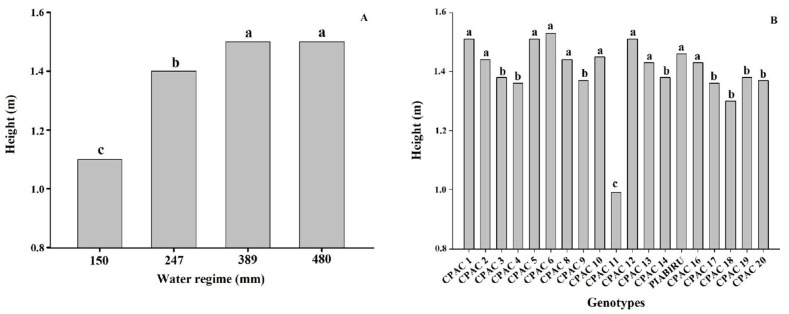
Height of different quinoa genotypes under 4 water regimes. (**A**) Effect of 4 water regimes in plant height in 20 quinoa genotypes. (**B**) Means followed by the same lowercase letter do not differ according to the Scott–Knott test at 5% probability.

**Table 1 plants-10-01591-t001:** Productivity (t ha^−1^) of 18 quinoa genotypes and BRS Piabiru under 4 water regimes.

Genotypes	Water Regime (mm)			
	480	389	247	150
CPAC1	7.82 aA	8.34 bA	4.06 bB	1.56 bC
CPAC2	8.32 aA	7.89 bA	3.94 cB	1.83 bC
CPAC3	8.02 aA	7.81 bB	6.25 aC	1.94 bD
CPAC4	8.40 aA	8.23 bA	5.21 aB	1.95 bC
CPAC5	8.17 aA	7.90 bA	5.22 aB	1.94 bC
CPAC6	8.50 aA	8.84 aA	5.68 aB	2.46 aC
CPAC8	8.64 aA	7.07 cB	4.50 aC	1.58 bD
CPAC9	8.21 aA	7.01 cB	5.75 aC	2.11 bD
CPAC10	8.56 aA	8.36 bA	5.12 aB	2.61 aC
CPAC11	5.66 bB	6.80 cA	5.40 aB	2.38 aC
CPAC12	8.57 aA	7.83 bA	5.82 aB	2.58 aC
CPAC13	8.85 aA	9.73 aA	4.17 bB	2.60 aC
CPAC14	9.21 aA	6.51 cB	3.88 bB	1.61 bB
BRS Piabiru	7.58 aA	8.14 bA	5.40 aB	1.84 bC
CPAC16	8.51 aA	7.46 cA	4.74 aB	2.33 aC
CPAC17	8.92 aA	8.44 bA	4.80 aB	3.64 aC
CPAC18	8.96 Aa	9.16 aA	4.75 aB	2.21 bC
CPAC19	7.71 aA	6.42 cB	2.53 cC	1.79 bC
CPAC20	7.97 aA	7.08 cA	3.93 bB	2.09 bC

Means followed by the same lowercase letter (column) or uppercase letter (line), do not differ according to the Scott–Knott test at a 5% probability.

**Table 2 plants-10-01591-t002:** Productivity per unit of applied water (kg ha^−1^mm^−1^) of 18 quinoa genotypes and BRS Piabiru under 4 water regimes.

Genotypes	Water Regime (mm)			
	480	389	247	150
CPAC1	17.18 aB	22.2 aA	16.43 cB	10.41 cC
CPAC2	15.89 aB	21.0 aA	13.02 dB	13.0 cB
CPAC3	18.59 aB	20.8 aB	25.29 aA	13.5 cD
CPAC4	16.89 aB	21.15 aA	21.1 bA	13.03 cC
CPAC5	16.9 aB	21.0 aA	23.9 bA	12.1 cC
CPAC6	17.7 aB	22.72 aA	23.0 aA	16.4 bB
CPAC8	18.01 aA	18.18 bA	18.23 cA	10.52 bB
CPAC9	17.73 aB	18.01 bB	23.3 aA	14.10 cC
CPAC10	17.83 aB	21.49 aA	20.6 bA	15.38 cB
CPAC11	11.1 bC	18.0 bB	21.89 bA	15.85 bA
CPAC12	18.0 aB	20.1 aB	23.56 aA	17.24 bD
CPAC13	19.83 aB	25.9 aA	19.1 bB	17.6 bB
CPAC14	12.69 aB0	16.3 bA	15.71 cA	10.1 cC
BRS Piabiru	15.79 aB	20.92 aA	21.88 bA	14.37 cB
CPAC16	17.74 aA	19.8 bA	19.18 bA	15.57 bA
CPAC17	18.58 aB	21.70 aA	19.44 bB	24.25 aA
CPAC18	18.66 aB	23.56 aA	16.89 cB	14.71 cC
CPAC19	16.08 aA	16.50 bA	11.6 dB	10.25 cB
CPAC20	17.82 aA	18.82 bA	15.94 cA	11.89 cB

Means followed by the same lowercase letter (column) or uppercase letter (line) do not differ according to the Scott–Knott test at a 5% probability.

**Table 3 plants-10-01591-t003:** Total flavonoid concentrations (mg/100 g) in grains of 18 quinoa genotypes and BRS Piabiru under 4 water regimes.

Genotypes	Water Regime (mm)			
	480	389	247	150
CPAC1	81.59 cB	101.62 dA	66.21 dC	88.34 dA
CPAC2	95.97 bA	97.23 dA	85.71 dA	92.21 dA
CPAC3	98.83 bA	51.54 eB	97.12 cA	113.25 cA
CPAC4	80.02 cB	98.31 dA	91.64 cA	96.23 dA
CPAC5	84.07 cB	103.0 dA	74.58 dB	86.80 dB
CPAC6	83.65 cC	115.23 cA	82.81 dC	100.05 dB
CPAC8	96.95 bB	110.72 dA	114.25 bA	93.58 dB
CPAC9	215.22 aB	171.32 aC	205.11 aB	226.02 aA
CPAC10	85.48 cC	110.0 dB	96.32 cC	210.81 bA
CPAC11	89.00 cB	118.59 cA	100.97 cB	113.25 cA
CPAC12	96.17 bB	120.98 cA	84.82 dB	112.51 cA
CPAC13	99.47 bA	111.33 dA	84.73 dB	102.33 dA
CPAC14	115.19 bA	119.74 cA	84.94 dB	104.71 cA
BRS Piabiru	104.15 bA	105.89 dA	101.98 cA	104.56 cA
CPAC16	105.09 bB	124.80 cA	99.48 cB	110.93 cB
CPAC17	115.73 bC	145.76 bA	97.05 cC	118.86 cB
CPAC18	88.27 cB	62.76 eC	94.25 cB	113.57 cA
CPAC19	109.63 bB	119.74 cA	102.59 cB	100.01 dB
CPAC20	119.93 bB	139.03 bA	114.40 bB	115.41 cB

Means followed by the same lowercase letter (column) or uppercase letter (line) do not differ according to the Scott–Knott test at a 5% probability.

**Table 4 plants-10-01591-t004:** Total anthocyanin concentrations (mg/100 g) in grains of 18 quinoa genotypes and BRS Piabiru under 4 water regimes.

Genotypes	Water Regime (mm)			
	480	389	247	150
CPAC1	0.80 cA	0.76 dA	0.76 cA	0.80 dA
CPAC2	0.63 dA	0.68 dA	0.62 dA	0.40 cB
CPAC3	0.63 dD	1.0 bB	0.83 cC	1.16 cA
CPAC4	0.59 dB	0.65 eB	0.59 dB	0.76 dA
CPAC5	0.74 cA	0.59 eB	0.54 dB	0.70 dA
CPAC6	0.60 dB	0.73 dA	0.65 dB	0.84 dA
CPAC8	0.58 dB	1.07 bA	0.55 dB	0.57 eB
CPAC9	1.72 aC	1.89 aB	1.44 aD	2.05 aA
CPAC10	1.16 bA	0.73 dB	0.76 cB	0.91 dB
CPAC11	0.61 dA	0.48 fA	0.54 dA	0.61 eA
CPAC12	0.55 dB	0.68 dA	0.50 dB	0.65 eB
CPAC13	0.66 dB	0.63 eB	0.50 dC	0.73 dA
CPAC14	0.83 cB	0.82 cB	0.68 cC	0.94 cA
BRS Piabiru	0.89 cB	1.08 bA	0.61 dD	0.74 dC
CPAC16	0.72 cA	0.65 eA	0.81 cA	0.80 dA
CPAC17	1.03 bB	0.86 cC	0.79 cC	1.18 bA
CPAC18	0.68 dA	0.61 eA	0.72 cA	0.60 eA
CPAC19	0.64 dA	0.63 eA	0.71 cA	0.66 eA
CPAC20	1.13 bA	0.99 bA	1.06 bA	1.05 cA

Means followed by the same lowercase letter (column) or uppercase letter (line) do not differ according to the Scott–Knott test at 5% probability.

**Table 5 plants-10-01591-t005:** Stomatal conductance (*g_s_*–mol m^−2^ s^−1^), transpiration rate (*E*–mmol m^−2^ s^−1^ Internal carbon (*C_i_*-μmol m^−2^ s^−1^), and photosynthetic rate (*A*-μmol m^−2^ s^−1^) of 4 quinoa genotypes (CPAC4, CPAC9, CPAC11, and Piabiru), under 4 water regimes (150, 247, 389, 480 mm).

Genotypes		Water Regime (mm)			
		150	247	389	480
***gs***	CPAC4	0.034 cD	0.235 bC	0.470 bB	0.583 bA
	CPAC11	0.040 cC	0.232 bB	0.358 cA	0.342 cA
	BRS Piabiru	0.058 aD	0.218 bC	0.516 abB	0.629 aA
	CPAC19	0.050 bC	0.329 aB	0.552 aA	0.594 abA
***E***	CPAC4	1.45 bcD	6.15 aC	8.96 aB	9.88 aA
	CPAC11	1.20 cD	4.87 bC	6.43 cA	5.67 cB
	BRS Piabiru	3.11 aD	4.76 bC	8.20 aB	9.46 aA
	CPAC19	1.71 bD	5.83 aC	7.27 bB	8.38 bA
***Ci***	CPAC 4	144.0 aC	195.3 aB	239.9 bA	241.2 bA
	CPAC11	112.6 bC	198.6 aB	253.6 aA	259.2 aA
	BRS Piabiru	129.0 abD	191.3 aC	257.0 aB	270.3 aA
	CPAC19	82.5 cC	198.4 aB	255.9 aA	256.9 abA
***A***	CPAC4	4.9 cD	19.0 cC	22.0 bB	28.1 bA
	CPAC11	5.9 bC	20.5 bcB	22.9 bA	20.0 cB
	BRS Piabiru	9.0 aD	22.4 bC	32.9 aB	35.4 aA
	CPAC19	8.9 aC	27.0 aB	32.9 aA	34.3 aA

Means followed by the same lowercase letter (column) or uppercase letter (line) do not differ according to the Tukey test at a 5% probability.

## Data Availability

Data is contained within the article and Appendix A.

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
