# Peer review of "Quinoa for the Brazilian Cerrado: Agronomic Characteristics of Elite Genotypes under Different Water Regimes"

_plants, 2021, doi:10.3390/plants10081591_

Round 1
Reviewer 1 Report
I found this manuscript particularly interesting. The authors study the effects of different irrigation regimes of Quinoa crops in a subtropical savannah of Brazil. The study area is often subjected to water stress, the authors propose different irrigation systems that improve the productivity of the plant and its physiology.
The manuscript is well written and proposes a new approach for cultivation in areas affected by drought for this reason I suggest to the authors these two recent manuscripts:
Aghajanlou, F. Rangeland Management and Ecological Adaptation Analysis Model for Astragalus curvirostris Boiss. Horticulturae 2021, 7, 67. doi: 10.3390/horticulturae7040067
Rad, S.V. Quantitative and Qualitative Evaluation of Sorghum bicolor L. under Intercropping with Legumes and Different Weed Control Methods. Horticulturae 2020, 6, 78. doi: 10.3390/horticulturae6040078
Author Response
The first reviewer requested the inclusion of two references.
Aghajanlou, F. Rangeland Management and Ecological Adaptation Analysis Model for Astragalus curvirostris Boiss. Horticulturae 2021, 7, 67. doi: 10.3390/horticulturae7040067
Rad, S.V. Quantitative and Qualitative Evaluation of Sorghum bicolor L. under Intercropping with Legumes and Different Weed Control Methods. Horticulturae 2020, 6, 78. doi: 10.3390/horticulturae6040078
Authors
The first reviewer requested the inclusion of two references.
The inclusion of two references in the article by the reviewer was suggested. We appreciate the suggestion to include these references, but they are not related to the submitted article.
Reviewer 2 Report
Review of the manuscript: Quinoa for the Brazilian Cerrado: Agronomic characteristics of elite genotypes under different water regimes
KEYWORDS: I strongly suggest authors to introduce more keywords. The usefulness of keywords is to make the article both more and more easily searchable visible after its publication through commonly used search engines.
Introduction: The introduction is interesting, but in my opinion, it does not fully cover the topic. Below are some suggestions on how to expand this section. Moreover, out of 12 cited items, some are older than 10 years. The authors refer to some very old literature (e.g. items 2 and 8). Can these items not be replaced with newer ones?
Materials and Methods: The quantification of total flavonoids and anthocyanins should be further described. Moreover, these determinations are based on very old methods (references 17 and 18). More accurate methods could have been used.
The topic is of interest and the experimental planning and execution are good. I believe that the work presented for review is of a high technical level. I am asking for a deeper description, taking into account my suggestions above, with post new items.
Author Response
The second reviewer requested some alterations in the paper. We did all of them, as follows:
Review of the manuscript: Quinoa for the Brazilian Cerrado: Agronomic characteristics of elite genotypes under different water regimes
KEYWORDS: I strongly suggest authors to introduce more keywords. The usefulness of keywords is to make the article both more and more easily searchable visible after its publication through commonly used search engines.
Authors
Three more keywords were included in the article: Drought; Canopy temperature; chlorophyll
Reviewer
Introduction: The introduction is interesting, but in my opinion, it does not fully cover the topic. Below are some suggestions on how to expand this section. Moreover, out of 12 cited items, some are older than 10 years. The authors refer to some very old literature (e.g. items 2 and 8). Can these items not be replaced with newer ones?
Authors
The oldest references were removed, except one of them (C. R. Spehar, "NOVAS CULTIVARES Quinoa BRS Piabiru : alternativa para diversificar," Pesq. agropec. bras, no. 1, pp. 889-893, 1998). We corrected the date of this paper. The correct date in 2001 and not 1998.
We kept this reference, as this is the first work done in Brazil with the quinoa culture, using the cultivar BRS Piabiru. It is a reference literature for Brazil.
Authors
The references were updated and new updated information on the physiology of quinoa under water stress was included.
Reviewer
Materials and Methods: The quantification of total flavonoids and anthocyanins should be further described. Moreover, these determinations are based on very old methods (references 17 and 18). More accurate methods could have been used.
Authors
The objective of this work was to determine the contents of total flavonoids and anthocyanins, and the method of refeernce 17 (in the old version of the paper, and now is number 18) is widely used for its simplicity, speed and accuracy. The work did not aim at a flavonoid profile to use a liquid chromatography method, for example.
Reviewer
The topic is of interest and the experimental planning and execution are good. I believe that the work presented for review is of a high technical level.
I am asking for a deeper description, taking into account my suggestions above, with post new items.
Authors
We detailed the method and also corrected the reference.
Reviewer 3 Report
The information shown in line 59-64, about flavonoids and anthocyanins in the quinoa plant, is not included in the results and their discussion. Furthermore, the relationship of these flavonoids and anthocyanins in the response to stress of the different genotypes of quinoa plantain is not clearly shown. It is recommended to carry out the determination in the leaves of the plant and not only in the quinoa grain.
The conclusion also did not mention how flavonoids and anthocyanins influence the mechanisms involved in stress tolerance in the different quinoa genotypes.
Author Response
Third Reviewer observations:
The information shown in line 59-64, about flavonoids and anthocyanins in the quinoa plant, is not included in the results and their discussion.
Authors
We corrected this part and agree with the reviewer. The new version of the text is below:
Quinoa (Chenopodium quinoa Willd.) is a pseudocereal rich in natural antioxidants, flavonoids and anthocyanins [6], [7], and these compounds may protect plants against biotic and abiotic stresses [8]. Water stress reduces crop height, stomatal conductance, plant biomass and yield [9], [10].
Reviewer
Furthermore, the relationship of these flavonoids and anthocyanins in the response to stress of the different genotypes of quinoa plantain is not clearly shown. It is recommended to carry out the determination in the leaves of the plant and not only in the quinoa grain.
Authors
The main objective was to measure the effect of stress on compounds related to functional quality and, therefore, it was measured in the grain that is the part consumed.
Reviewer
The conclusion also did not mention how flavonoids and anthocyanins influence the mechanisms involved in stress tolerance in the different quinoa genotypes.
Authors:
The main objective was not to measure the effect of compounds linked to functional quality on stress but how the stress affects anthocyanin and flavonoids.
However, we included informations about anthocyanin and flavonoids in the conclusions:
“The accumulation of flavonoids and anthocyanins in the grains was more influenced by quinoa genotypes than by the water regimes”
We corrected the bibliography according to the journal's rules.
We have refined the results in Tables 3 and 4 (lines 328-333).